# Energy Consumption Optimization Model of Multi-Type Bus Operating Organization Based on Time-Space Network

**Yuhuan Liu [1]** , **Enjian Yao [2],\* and Shasha Liu [1]**

[1]  School of Traffic and Transportation, Beijing Jiaotong University, Beijing 100044, China
[2]  Key Laboratory of Transport Industry of Big Data Application Technologies for Comprehensive Transport, Beijing Jiaotong University, Beijing 100044, China
\*  Correspondence: enjyao@bjtu.edu.cn



**Featured Application: The findings obtained from this study have implications for application of the pure electric bus.**

**Abstract:** As a new type of green bus, the pure electric bus has obvious advantages in energy consumption and emission reduction compared with the traditional fuel bus. However, the pure electric bus has a mileage range constraint and the amount of charging infrastructure cannot meet the demand, which makes the scheduling of the electric bus driving plans more complicated. Meanwhile, many routes are operated with mixing pure electric buses and traditional fuel buses. As mentioned above, we focus on the operating organization problem with the multi-type bus (pure electric buses and traditional fuel buses), aiming to provide guidance for future application of electric buses. We take minimizing the energy consumption of vehicles, the waiting and traveling time of passengers as the objectives, while considering the constraints of vehicle full load limitation, minimal departure interval, mileage range and charging time window. The energy consumption based multi-type bus operating organization model was formulated, along with the heuristic algorithm to solve it. Then, a case study in Beijing was performed. The results showed that, the optimal mixing ratio of electric bus and fuel bus vary according to the variation of passenger flow. In general, each fuel bus could be replaced by two pure electric buses. Moreover, in the transition process of energy structure in public transport, the vehicle scale keeps increasing. The parking yard capacity and the amount of charging facilities are supposed to be further expanded.

**Keywords:** pure electric buses; multi-type bus operating organization; time-space network; energy consumption; public transport

---

## 1. Introduction

In China, with the rising amount of motor vehicles, the energy consumption is aggravating. At the same time, the automobile exhausted emissions bring severe air pollution, and the energy and environmental problems caused by transportation vehicles are becoming increasingly serious. Driven by the oil crisis and environmental pressure, the government and enterprises are actively promoting the development of pure electric buses. According to statistics, the cost of energy consumption accounts for 30% to 40% of the operating costs of bus operation enterprises. In addition, under the background of the national policy of energy conservation and emission reduction, many bus operators are confronted the targets of the total energy consumption control. Once the target is not completed, they will face penalties. Pure electric buses have the advantages of high energy efficiency and low pollution and

they are rapidly replacing traditional fuel buses, accounting for a gradually increasing proportion in the bus fleet. However, pure electric buses have an insufficient driving mileage range, long charging time and the amount of charging infrastructure is not enough, which adds the complexity of bus operation organization. Under the background of application of the pure electric buses, it is the main support to realize the large-scale pure electric bus operation by strengthening the study on the scheduling driving plan based on the constraints of the pure electric bus mileage range and charging time window, ensuring the vehicles to complete the daily operation tasks with maximum reduction on energy consumption under the condition of pure electric buses and fuel buses mixing operation.

In the field of the bus operating organization considering driving ranges constraint, Bodin L et al. [1] described the problem, and focused on the constraint of the mileage limit of a vehicle without returning to the yard to replenish energy after the departure. Freling. R et al. [2] discussed the driving ranges constraint as well. This vehicle operating organization method is based on a single parking yard, solving the problem by considering the battery life constraint in the single trip of entering and leaving parking yard, without taking into account the situation that the bus drives back to the yard halfway to charge and continue to conduct the tasks, which makes the model not suitable for the problem considering fuel consumption constraints. Haghani et al. [3,4] studied the problem of the regional bus operating organization based on the driving ranges battery life constraint. The paper classified the buses into three shifts, morning, noon and evening, and divided them into the yard matching shifts and the route matching shifts. The model reduced the size of the problem by about 40% without reducing any feasible solutions. The application showed that the research could effectively increase the vehicle operating time and cut the operation cost of bus operators. However, it adopted the post-check strategy, which limited the optimization of the solution to some extent. Ali Haghani et al. [5] constructed a mathematical model for the multi-depot scheduling with path time constraints, and used an exact algorithm and two heuristic algorithms to solve the model. Amar Oukil [6], Guy Desaulners [7] and M.A. Forbes [8] also studied the multi-depot vehicle scheduling problem, but they used the column generation algorithm and exact algorithm, respectively to solve the problem. Ali Haghani [4] compared one multi-depot vehicle scheduling model with two single-depot vehicle scheduling models. Natalia Kliewer [9] and Pablo C. Guedes [10] modeled the multi-depot bus scheduling problem based on the spatio-temporal network.

In terms of solving the operating organization optimization model, hyper-heuristic and hybrid algorithms have attracted much attention in recent years. Pepin et al. [11] compared a variety of hybrid heuristic methods for vehicle scheduling, finding that the heuristic method based on column generation had the best solution quality yet with a long solution time, while the heuristic method based on large-scale neighborhood search was fast and of good quality. Among the soft computing method applied to the vehicle scheduling, VAMPIRES [12,13] was one of the most successful early heuristic methods, having the similar main ideas to 2-opt algorithms. It had successfully solved hundreds of actual public transport vehicle scheduling problems, and later was replaced by the the BOOST object-oriented software system. Wren and Kwan [14] reported the application of the system in a British bus company. In the past ten years, more researches focused on the realistic constraints or characteristics, resulting in the emergence of a series of hyper-heuristic and hybrid approaches. For example, Shen [15] developed a vehicle scheduling method based on the tabu search by applying the 2-opt algorithm. Eliiyi et al. [16] proposed six types of meta-heuristic methods to solve vehicle scheduling problems with multiple vehicle types and continuous driving time constraints.

Since the 1990s, the focus of the problem research has been on solving the exact solution of the problem. Some scholars had proposed the precise branch-bound method and the precise column generation method. In order to reduce the scale of the problem, the time-space network was introduced into the multi-yard vehicle operating organization. Kliewer et al. [17] applied the concept of the time-space network to the multi-station vehicle operating organization problem for the first time and explained the method of cutting empty-drive arcs in the network. In addition, it introduced the method and steps of reducing the network scale, proposed the multi-commodity network flow model of the

multi-type multi-yard bus driving plan scheduling problem based on the time-space network, and solved an example problem with a large-scale vehicle operation organization by using the standard mathematical optimization software CPLEX. Naumaim et al. [18] provided a multi-commodity network flow model based on the time-space network and proposed a stochastic programming algorithm to solve the model. It was of great significance to simplify the problem, establish the model and solve the problem. He Di et al. [19] analyzed the connotation of bus regional dispatching and distribution planning problem, constructed a model of bus regional dispatching and distribution planning based on the space-time network, and verified the model and algorithm through an example. Yang Yang et al. [20] transformed the planning problem of the electric bus into a directed network. Bodin L et al. [21] and others put forward the idea of two-stage heuristics to solve the problem of multi-station traffic planning for the first time. Dell Amico et al. [22] took the minimum number of vehicles required as the optimization objective and used the heuristic algorithm to solve the problem in stages. Freling, R et al. [23] considered decision variables describing the connection between the vehicles and assigning vehicles to each station. Laurent [24] solved the problem of the multi-station traffic planning based on the iterative local search algorithm, and analyzed 30 cases.

At present, the research on the bus driving region dispatching is not very mature. Wei Ming et al. [25] established a mixed integer programming model with time windows, aiming at the minimum number of vehicles, vehicle waiting time and empty driving time, taking into account factors such as yard capacity, allowable vehicle refueling and task reliability of each vehicle. On the basis of describing the vehicle scheduling problem with time windows, multiple yard and multiple vehicle types, Yang Yuanfeng [26] established a mathematical model and proposed a simulated annealing genetic algorithm to solve the problem. The strong climbing performance of simulated annealing algorithm could avoid the "premature" of the genetic algorithm and improve the convergence speed of the algorithm. Zou Ying [27] took the regional scheduling of multiple bus lines as the service object, established a bus driving plan model with the objective of minimizing the passengers waiting cost, in-bus cost and the cost of the bus companies and proposed the solution idea of "allocate shifts by-line, optimize to form a network". Li Jun [28] proposed a heuristic algorithm based on dispatching after analyzing the vehicle scheduling problem with time windows. The algorithm defined two kinds of dispatching costs, designed a method to arrange routes in the dispatching process, and conducted case verification. Later, Li Jun [29] proposed a heuristic algorithm based on network optimization, which transformed the vehicle scheduling problem with time windows into several vehicle scheduling problems with a certain start time, then, used the minimum cost and the maximum flow algorithm to solve the vehicle scheduling problem with a certain start time.

To sum up, there are few researches on the optimization theory of regional operating organization currently. From the perspective of the network description, most of them emphasize the fixed allotment relationship between vehicles and vehicle yard. The researches without the fixed allotment relationship between the vehicles and vehicle yard are mainly found in logistics distribution, while fewer studies are in the field of the bus operating organization. The existing researches on the vehicle driving plan model are too simple, which greatly simplify the real operating environment, and the results are difficult to adapt to the complex work. Additionally, the current researches are hardly carried out under the background of the mixing operation with multiple vehicle types such as electric buses and fuel buses, and there are no regional bus driving plan models considering both the mileage range and charging time constraints. Some literature took the battery life into account, but they only discussed the battery life constraint within one shift and did not consider the condition that the vehicle drives back to the yard for charging and continues the tasks. Furthermore, there is little research on bus energy consumption. Therefore, we conduct the research on the regional pure electric bus driving plan with the objectives of energy consumption and bus service level, considering the pure electric bus charging constraints and mileage range constraints, and modify the algorithm to better support the application of pure electric buses in the real operation.

## 2. The Establishment of Time-Space Network

### 2.1. Establish the Time-Space Network

The time-space network is composed of a large number of nodes and various arcs. While establishing the network, the simplification of empty-driving arcs should be considered at the generation stage. We do not distinguish the vehicles belonging to different parking yards, and there is no fixed allotment relationship between vehicles and parking yards. As long as the number of vehicles in the yard is unchanged, there is no necessity that the vehicles must return to the original departing parking yard. Therefore, the yards can be regarded as general stations, and used to express the connection between yards and other stations in the same network. The processes of establishing multi-type pure electric buses system time-space network with the characteristic of recharging requirement are as follows:

1. Generate task nodes and task arcs. Generate the task start node and task end node according to the route timetable, and each node has its own time and space attributes. Assuming that there are $N$ task vehicles, $S_i$ is the departure site of number $i$; $e_i$ is the arrival site of number $i$; $d_i$ is the departure time of number $i$; $a_i$ is the arrival time of number $i$. According to the above process, the start node $t(S_i, d_i)$ can be formed, and the start node of the task can be arranged in chronological order to form a set of task start nodes T; the end node $e(e_i, a_i)$ can be arranged in chronological order to form a set of end-of-task nodes E. Task arc: $(t, e)$, $t \in T, e \in E$. The set of task arcs is $A_{task}$. Connect the task start node with the corresponding task end node to generate the task arc.

2. Generate the parking lot node, exit arc, and entry arc. According to the task start node set, the task end node set and the empty-driving time between each station and parking yard, set up the corresponding exit node and entry node (collectively regarded as the parking yard node). Then set up the exit arc connecting the exit node and task start node, and the entry arc connecting the task end node and entry node.

3. Generate the waiting arc. Sort by stations first, then arrange them in time order and generate a collection of all the nodes.

4. Generate the empty-driving arc. Take out the task end node set and generate a DHE set according to the sequence of the task end time from the smallest to the largest. Sort the task start nodes according to the stations firstly, and then generate a T set according to the sequence of time from the smallest to the largest. Initialize the empty-driving arc.

### 2.2. The Representation of Vehicle Operating States

In the processes of optimizing the energy consumption of the multi-type buses operating organization, it is assumed that when there are no pure electric buses that can operate, the fuel buses will be arranged. Compared with the charging time of electric buses, the refueling time of the fuel buses is negligible, which can be assumed that the fuel buses can run unlimitedly. Assuming that if the energy of a pure electric bus is not enough to run the next task, it cannot charge in the halfway, instead, it needs to return to the yard to charge. Therefore, after the bus completes a certain task, it would have one of the following states. (1) Get into the parking yard through the entry arc and wait for the next task, as shown in Figure 1a; or considering the mileage range of the pure electric bus, if it has reached the maximum mileage, it has to get into the yard through the entry arc to recharge, as shown in Figure 1b. (2) The bus departs from the terminal station either directly or after a period of waiting, conduct the next task departing from this terminal station, which is shown in Figure 1c. (3) The bus arrives at the departure station of another task through an empty-driving arc to operate the next task. There is a certain waiting time between the termination of this task and the departure of the next task, which indicates there may be some empty-driving arcs and a certain number of waiting arcs between running the two tasks, which are shown in Figure 1d.

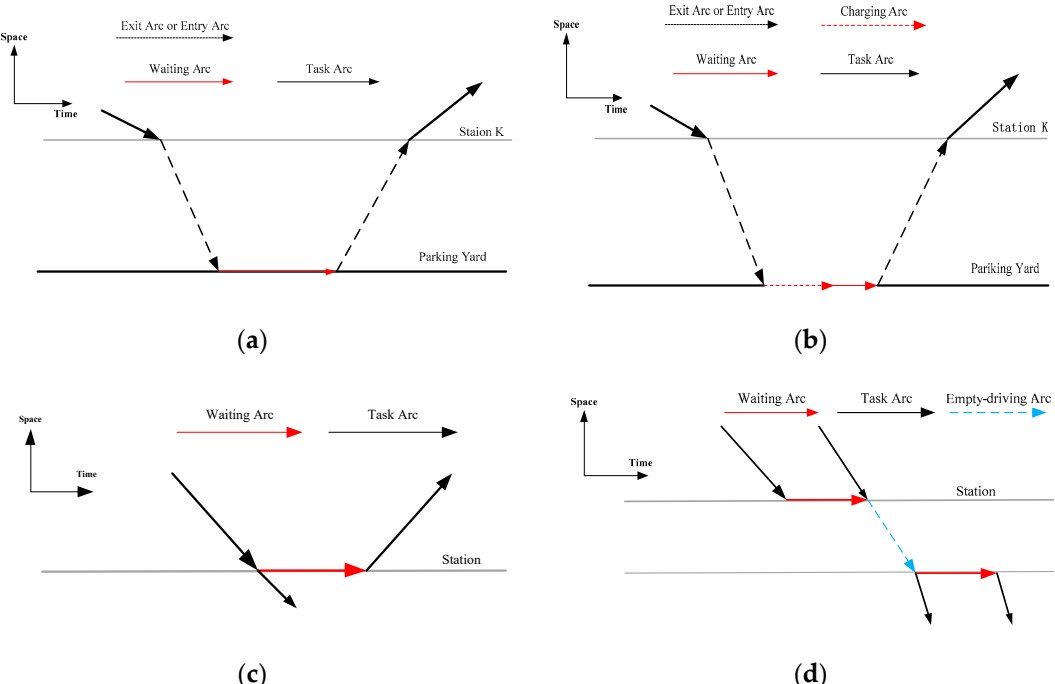

**Figure 1.** The representation of vehicle operating states in the time-space network. (**a**) The bus returns to the parking yard to wait for next task; (**b**) the pure electric bus gets into the yard to charge; (**c**) the connection of the two tasks at the same site; (**d**) the bus runs two tasks connecting with the empty running arcs.

## 3. Multi-Type Bus Operating Organization Energy Consumption Optimization Model

### 3.1. Problem Description

Compared with the operating organization of traditional buses, the multi-type bus operating organization has the constraints of both pure electric buses mileage range and charging time windows. In addition, in the context of "energy conservation and emission reduction", we not only consider the completion of operational tasks, but also take the reduction of energy consumption into consideration. Therefore, the bus operating organization studied in this paper is similar to the vehicle scheduling problem with the constraint of mileage range, charging time and the consideration of energy consumption optimization. At the same time, we study the regional bus driving mode, and the track of vehicles changes from a single route to several routes. The relationship among vehicles, stations, bus routes and task shifts become extremely complicated, making it more difficult to organize the operation of buses.

### 3.2. Model Establishment

#### 3.2.1. Model Assumption

In the process of replacing fuel buses with electric buses, the condition that fuel buses and electric buses are mixed operated exists. Based on this background, we establish an operating organization energy consumption optimization model with the following assumptions: (1) The bus driving plan is scheduled with the unit of day, and it takes a minute as the smallest unit of time; (2) according to the data collected from the Beijing Bus Group, the maximum mileage for the 12-meter-long electric buses is 133 km and 117 km for the 18-meter-long electric buses. The buses powered by fuel and natural gases have unlimited mileage; (3) the quick charging time for electric buses is 30 min; (4) all the buses can run the task on time, and the model does not consider the condition of delay.

### 3.2.2. Objective Function

In the process of optimizing the multi-type bus operating plan while controlling the total energy consumption, we consider the service level of the bus, which includes the comfort indicators (waiting time, full load rate), convenience indicators (transfer distance, station coverage rate) and economic indicators (fare). However, for fixed bus lines, the transfer distance and station coverage indicators have been determined, and the ticket fare has no relation to the departure frequency, so the service level indicators considered in the paper mainly contains the waiting time and full load rate. The time cost of passengers waiting for a bus is mainly related to the time of waiting, and the full load rate is associated with the time cost of the passengers' in-bus time, because passengers have different perceptions of the travel time for the same travel distance under different full load rates. Therefore, we establish a multi-objective bus operating organization energy consumption optimization model. The objectives include the total energy consumption, passengers waiting time and passenger's in-bus time. By transforming different objectives into cost, multiple objectives can be converted into a single optimization objective with a unified unit of measurement, which can reduce the complexity and difficulty of solving the model.

The minimum total energy consumption $E$ can be calculated as the sum of the product of vehicle per unit energy consumption and driving mileage for each task, which is shown as Formula (1).

$$\min E = \sum_{p \in \Omega} \sum_{k \in K} \sum_{m \in M} e_k L_p \beta_{mp}^k \theta_p \tag{1}$$

$\Omega$ is the set of all the feasible bus order chain solutions. $P$ represents the bus order chain. $M$ is the set of buses. $m$ represents the bus. $K$ represents the vehicle type set. $e_k$ is the unit (per kilometer) energy consumption of type k vehicles. $L_P$ is the mileage of the bus order chain $P$. $\beta_{mp}^k$ is a variable ranging from zero to one, representing whether the bus order chain $P$ is executed by the bus $m$ of the vehicle type $k$. If it is, $\beta_{mp}^k$ is one, otherwise, $\beta_{mp}^k$ is zero. $\theta_p$ is a variable ranging from zero to one, indicating whether the bus order chain $P$ is in the feasible solution. If it is, then $\theta_p$ is one, otherwise $\theta_p$ is zero. $P = \{1, 2, 3, \ldots, i, n\}$ is a set representing the sequence of the exit arc, task arc, waiting arc, empty-driving arc, and entry are executed by a bus departing from a certain parking yard.

$C_1$ is the minimum passengers' waiting time. Since the passengers' waiting time cost is mainly decided by the waiting time, and the waiting time is associated with the departure frequency of the bus, the relationship between the departure frequency and the passengers' waiting time can be described as follows. Assuming that the law of the buses and the passenger arriving at the bus stations are subject to the Poisson Distribution and uniform distribution, respectively. The passengers' waiting time on average is the half of the departure interval and the bus lines' departure interval in unit time is equal to the reciprocal of departure frequency. The passengers' waiting time cost is inversely proportional to the departure frequency, and directly proportional to the number of people boarding the bus in the stations, so the calculation of the waiting time is shown in Formula (2).

$$\min C_1 = \frac{1}{2f} \sum_n \chi^n \tag{2}$$

$f$ is the departure frequency of the research bus line in the unit time period, with the unit of times/h. $\chi^n$ is the number of passengers boarding the bus at station $n$ in a unit time period, with the unit of persons.

$C_2$ is the minimum passengers in-bus time. The passengers' perception of the in-bus time cost is mainly influenced by the degree of the in-bus crowding (full load rate), which is determined by the departure frequency to some extent. The in-bus time cost of the same travel distance perceived by passengers is various under different crowding degrees, which is mainly affected by the in-bus time perception coefficient. Therefore, we consider that the passenger's in-bus time cost is mainly

determined by the cross-sectional passenger flow per unit period, travel time and passenger's in-bus perception coefficient, as shown in Formula (3).

$$\min C_2 = \min \sum_{k} \sum_{n} x_{n,n+1} \frac{l_{n,n+1}}{v_k} F_k(x) \tag{3}$$

$$F_k(x) = 1 + \beta_{\min} \left( \frac{x_{n,n+1}}{f \cdot N_k} \right)^{\beta_{\max}}$$

$x_{n,n+1}$ is the cross-sectional passenger flow between station $n$ and station $n + 1$ in a unit time period, with the unit of person. $l_{n,n+1}$ is the operation distance of the research route between station $n$ and station $n + 1$, with the unit of km. $v_k$ is the average velocity of the type $k$ bus. $F_k(x)$ is the passenger perception coefficient of the type $k$ bus when the cross-sectional passenger flow between station $n$ and station $n + 1$ is $x$. $N_k$ is the specified passenger capacity, with the unit of persons. $g$ represents the time period. $\beta_{\min}$ is the minimum allowable full load rate of the research route, $\beta_{\max}$ is the maximum allowable full load rate of the research route.

Therefore, by converting the multiple objectives into a single optimization objective, the multi-objective operating organization energy consumption optimization model can be represented as Formula (4).

$$\begin{aligned} \min z &= \min(\omega_1 E + \omega_2 C_1 + \omega_3 C_2) \\ &= \min(\omega_1 \sum_{p \in \Omega} \sum_{k \in K} \sum_{m \in M} e_k L_p \beta_{mp}^k \theta_p + \omega_2 \frac{1}{2f} \sum_{n} \chi^n + \omega_3 \sum_{k} \sum_{n} x_{n,n+1} \frac{l_{n,n+1}}{v_k} F_k(x)) \end{aligned} \tag{4}$$

$\omega_1, \omega_2, \omega_3$ is the cost converting coefficient.

### 3.2.3. Constraint Conditions

To ensure the original service level of the route, meeting the accurate matching between the transport capacity in the processes of replacing fuel buses with electric buses, while considering the mileage and charging time of electric buses, we establish constraints conditions from the aspect of the bus operating service level including the vehicle full load capacity, departure interval, road capacity, mileage of electric buses and charging time. The specific conditions are as follows:

1. The constraints of the full load rate

$$\beta_{\min} \leq \beta_{gh} \leq \beta_{\max} \tag{5}$$

$$\beta_{gh} = \frac{q_{gh}}{N_g C} = \frac{q_{gh} d_g}{N_g H_g}$$

In the formula, $\beta_{gh}$ is the cross-section full load rate of the $h$ section in a $g$ time period of the research route, $q_{gh}$ is the cross-section passenger flow of the $h$ section in a $g$ time period of the research route, $N_g$ is the number of passengers on board of the $h$ section in a $g$ time period of the research route, $d_g$ is the departure interval in a $g$ time period of the research route, with the unit of minute, $C$ is number of vehicles that passed the section $h$ in a $g$ time period, $H_g$ is the duration of the $g$ period, with the unit of minute, $N_0$ is the standard passenger capacity of the pure electric buses.

2. The constraint of departure interval

If the departure interval is too short, the energy consumption will increase that will result in the resource waste. If the departure interval is too long, the waiting time of passengers will be longer, and the full load rate will increase, then the service level decreases. Therefore, we comprehensively consider the acceptability of enterprises and passengers to the departure interval and refer to the Beijing bus departure interval which is less than 5 min in the peaking period and less than 15 min in

the low peak period. We widen the constraints of departure frequency within a certain range and set the following constraint conditions:

$$4 \leq f \leq 60 \tag{6}$$

3. The constraints of mileage and charging time windows

$$\sum_{p \in \Omega} a_{ip} \theta_p = 1 \; i \in A_{task} \tag{7}$$

$\alpha_{ip}$ is a variable either zero or one, indicating whether the bus order chain $P$ contains the arc $i$. If it does, $\alpha_{ip}$ equals to one, or $\alpha_{ip}$ equals to zero.

$$\sum_{p \in \Omega} s_{dp} \theta_p = \sum_{p \in \Omega} e_{dp} \theta_p \leq capacityd \quad \forall d \in D \tag{8}$$

$s_{dp}$ is a variable either zero or one, representing whether the bus order chain $P$ departs from the parking lot D or not. $e_{dp}$ is a variable either zero or one, representing whether the bus order chain $P$ stops at the parking lot d. $D$ represents the parking yard set. *capacityd* is the capacity of the parking yard $\forall d \in D$. $V_d^k$ is the vehicle set of type $k \in K$ departing from the parking yard $d \in D$.

$$\sum_i l_i a_{ip} \leq DD_k \forall p \in \Omega \quad i \in A_{task} \quad j \in A \tag{9}$$

$DD_k$ represents the maximum mileage of type $k \in K$ bus. $l_i$ is the mileage of arc $i$, and if the arc $i$ is the waiting arc, $l_i = 0$, otherwise, it equals the distance between the two stations.

$$a_{ip} arrtime_i \leq a_{jp} deptime_j \; i < j, \forall p \in \Omega \tag{10}$$

$arrtime_i$ represents the arrival time of arc $i$. $deptime_i$ represents the departing time of arc $i$.

$$s_{dp} e_{dp} y_i \theta_p \in \{0,1\} \; p \in \Omega, d \in D, i \in A_{task} \tag{11}$$

$y_i$ is the variable ranging either zero or one, indicating whether the bus returns to the parking yard to charge after completing the task $i$ (assuming that the bus can only be charged at the yard). If the bus needs to be charged, $y_i$ equals to one, and the corresponding entry time is $\alpha_i + t_{ei} + ST_k$. $ST_k$ is the charging time of the bus, which is 30 min in the paper.

In summary, we establish a bus driving plan optimization model considering the service level with the objective function (4) and constraint conditions from formulas (5) to (11). The constraint condition (7) means every task can only be executed once by one bus. The left-hand side of the constraint condition Formula (8) represents the number of bus order chains departing from parking yard d, and the right-hand side represents the number of bus order chains finally returning to the parking yard, and both of them should not exceed the total number of vehicles in the yard d. Formula (9) means the driving distance should be less than the maximum driving distance if the bus has not been supplied with energy. Formula (10) represents the time connection constraints between the arcs. Constraints condition (11) means the related variables ranging from zero to one.

## 4. Research on Heuristic Optimization Algorithm

In this paper, the energy consumption optimization problem of multi-type bus operating organizations is abstracted into a set segmentation problem, which can be described as the problem that which vehicle completes task shifts in turn. The integer coding method based on vehicles is adopted for chromosome coding, which can be simply expressed as [1231312...]. The coding represents that task one, four and six are completed by the first bus, task two and seven are completed by the second bus, and task three and five are completed by the third bus.

*4.1. The Production of Initial Population in Heuristic Algorithm*

To solve the model, the paper sets the heuristic algorithm with the following steps:

1.  Arrange the tasks in ascending order according to the task departure time to form T, and the task information includes the task sequence number, route number, departure time, arrival time, departure station, arrival station, travel time, route mileage, the nearest parking yard to the arrival station and the distance between them.
2.  Set up the vehicle set in each parking yard, including the parking yard number and its capacity.
3.  Assign one bus for the first task (the bus has not executed the task, which means the last task shift is empty), randomly extract an electric bus from the parking yard to execute, and record in the following order: The parking yard which the bus belongs to, the remaining mileage of the bus, the sequence number of the completed shift. Additionally, record the driving path of the current bus: The departure parking yard and the task sequence number.
4.  For the rest of task *i*:

Look for whether there are any already used electric buses that can still execute the task i, which should meet the following two requirements: 1. Remaining mileage subtracts possible empty-driving mileage is greater than task mileage adds the mileage returning to the nearest parking yard; 2. the time when the last task was completed adds possible empty-driving time is less than the start time of the task adds 30 min (the charging time).

According to the following rules, relate the bus to the task:

1.  Give priority to the buses meeting both requirements;
2.  Give priority to the buses meeting the second requirement but not the first. In this case, the bus can return to the parking yard to charge, making it meet the first requirement. Record the nearest charging yard and update the bus information and the driving path information.
3.  If all the requirements cannot be satisfied, the fuel buses will be used.

Look for the last task of each vehicle currently, assign the task *i* to the vehicle which has the smallest time difference from task *i*. Determine whether it is necessary to return to the parking yard to link up the tasks and update the vehicle information. All the individuals achieved based on the above heuristic algorithm are all the feasible solutions to the problem.

*4.2. Design of Fitness Function*

In the process of generating the initial solution, that is, the generation of the running plan of each vehicle, the corresponding task mileage, empty driving mileage, station or waiting time in the yard of the vehicle are recorded. In this paper, ObjV $= \sum_{p \in \Omega} c_p \theta_p$ is taken as the basis of the individual fitness evaluation, and the advantages and disadvantages of individuals are evaluated. The chromosomes that do not meet the constraints are eliminated. Calculate the objective function value of each individual and sort it. The fitness value of each individual is calculated according to its position in the arrangement. The fitness function of individual *i* is as follows:

$$g(i) = 2 - sp + 2(sp - 1)\left[\frac{pos - 1}{N - 1}\right] \tag{12}$$

In the formula, *sp* represents the selected pressure difference which is the difference between fitness values of assigned individuals, with a default value of two; *N* represents the size of the population; *Pos* represents its position in the arrangement.

*4.3. Genetic Manipulation*

Through selection, crossover and mutation to achieve the genetic manipulation, the specific process is as follows.

1. Selection. We adopt the roulette strategy RWS as the selection method. Since the probability of each individual being selected is proportional to its fitness function value. The probability of being selected is as follows:

$$p(i) = \frac{g(i)}{\sum_{i=1}^{N} g(i)} \tag{13}$$

On the basis of knowing the probability of each individual being selected, the selected individual is randomly determined. The specific steps are shown in Figure 2.

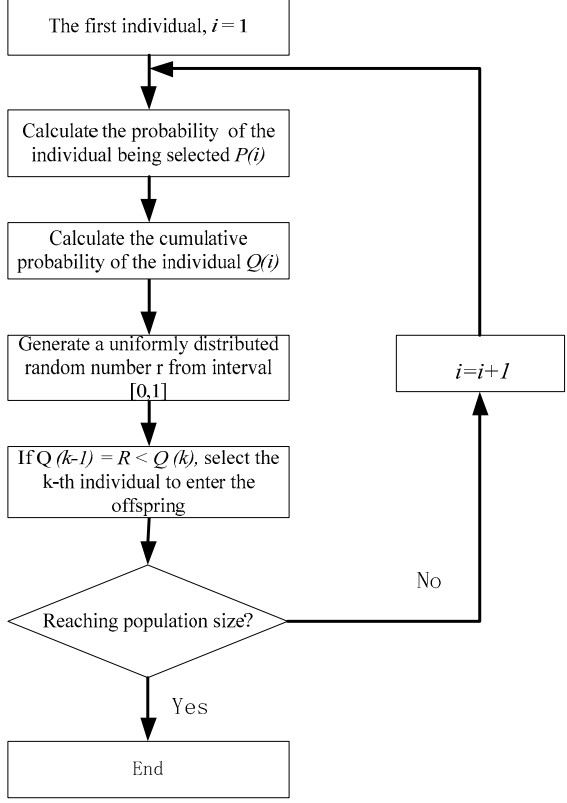

**Figure 2.** Individual selection step.

2. Crossover. The single point crossover is used. To adapt to the multi-constraint model proposed above, the following ideas are designed in the crossover operation.

    (a) Achieve the population from the selection operation for chromosome pairing;
    (b) According to the crossover probability, determine whether to carry out the crossover operation. If so, move to the next step, otherwise, move to the next species.
    (c) Search for feasible intersections. Decode to achieve the task shifts of each vehicle in the chromosome, randomly select two vehicles, and determine whether there is an intersection. If yes, conduct the crossover. Otherwise, pass it on to the next population until the two parents are inherited.
    (d) Move the chromosomes into the next population, determine whether the population size has been reached. If yes, terminate the crossover operation. Otherwise, repeat the above steps.

3. Variation. The variation ranges from 0.0001 to 0.1. Randomly select a certain task of a certain vehicle in the chromosome. Delete the task and the insert it into other vehicles at random, then determine whether the mileage range constraint can be satisfied. If it is feasible, then insert the task. Otherwise, continue to search for the vehicles that can be inserted. If no suitable insertion

position can be found in the existing vehicles, a new vehicle will be assigned to perform the corresponding task.

### 4.4. Design of Termination Principle

Before reaching the maximum number of iterations, determine whether the average fitness of successive generations has not changed, or the variation is less than a threshold. If so, the iterative process of the algorithm converges and the algorithm ends. Before reaching the maximum number of iterations, it is judged whether the average fitness of successive generations is unchanged or the change value is less than a minimum threshold. If so, the iteration process of the algorithm converges and the algorithm ends. Otherwise, if the maximum number of iterations has been reached, the new generation population obtained through selection, crossover and mutation will replace the previous generation population.

### 4.5. Flow Diagram of the Algorithm

The flow chart describing the established driving plan model is shown in Figure 3.

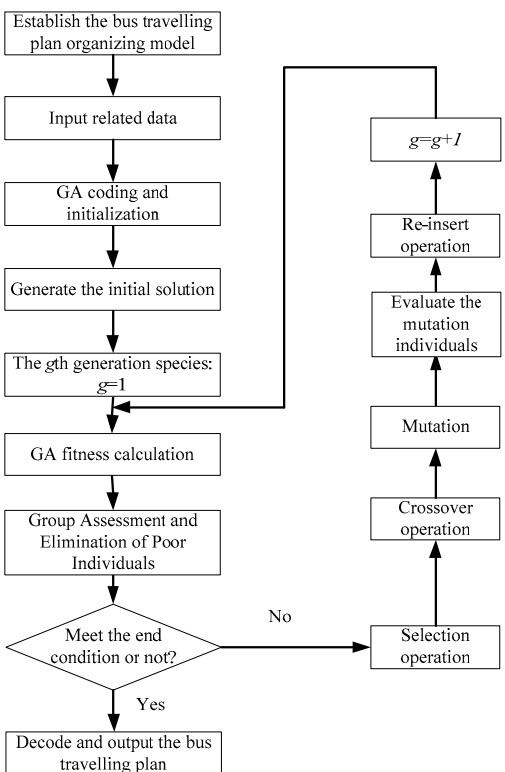

**Figure 3.** Flow diagram of the algorithm.

1.  Input the parameter of the genetic algorithm: Species size, species generation, crossover probability, mutation probability, and the generation gap.
2.  Input the task information, parking yard information, empty driving distance, average velocity, vehicle information and maximum mileage of pure electric buses. To make the calculation easier, the task information includes the task sequence number, route number, departure time, arrival time, departure station, arrival station, traveling time and route mileage. The parking yard information includes the parking yard number and the number of parking space in the yard. The vehicle information includes the vehicle number, parking yard number which the vehicle belongs to, remaining mileage, completion time of the last task, the sequence number of the last task, the information that whether the vehicle can execute the current task, the parking lot

number which the vehicle returns to, empty driving mileage, number of charge cycles and the vehicle energy consumption.

3. Input the coefficient of the vehicle energy consumption, the coefficient of passengers waiting cost and coefficient of passengers in-bus cost.

## 5. Result and Discussion

### 5.1. Explanation of Basic Information

Taking a certain bus system in Beijing as an example, we study multi-type buses scheduling and analyze an algorithm calculation example. The example contains six bus routes (Route 405, Route 415, Route 538, Route 1, Route 322 and Route 496), 12 timetables, 1040 tasks in total and seven bus parking yards. The parking yard information is shown in Table 1. The number of parking space in Table 1 is one of the constraints which is calculated by Formula (8).

**Table 1.** Parking yard information.

| Number of Parking Yard | Name of Parking Yard | Number of Parking Space |
|---|---|---|
| 1 | Si Hui Junction Station Parking Yard | 50 |
| 2 | Sun He Bus Parking Yard | 35 |
| 3 | Hui Zhong Li Parking Yard | 25 |
| 4 | National Stadium East Parking Yard | 25 |
| 5 | Lao Shan Bus Parking Yard | 25 |
| 6 | Gu Cheng West Bridge Bus Parking Yard | 25 |
| 7 | Kang Jing Nan Li Parking Yard | 25 |

The vehicle information is shown in Table 2. The energy consumption per 100 km is applied in the objective function (1). The specified passenger volume is one of the correlative conditions of the full load rate constraint Formula (5).

**Table 2.** Vehicle information.

| Type of Fuel | Length of Bus(m) | The Sequence Number of Vehicle Type | Energy Consumption per 100 km (Standard Coal) | Specified Passenger Capacity (Person) |
|---|---|---|---|---|
| **Electricity** | 18 | V1 | 55.76 | 130 |
| **Diesel** | 16 | V2 | 71.66 | 140 |
| **Natural Gas** | 16 | V3 | 75.47 | 164 |

The bus route information is shown in Table 3. The route mileage is the key index for calculating energy consumption of the objective function (1).

**Table 3.** Bus route information.

| Bus Route | Departure Parking Yard | Terminal Parking Yard | Route Mileage (km) |
|---|---|---|---|
| 405 | 1 | 2 | 22.31 |
| 415 | 3 | 2 | 17.69 |
| 538 | 4 | 2 | 19.76 |
| 1 | 1 | 5 | 24.84 |
| 322 | 1 | 6 | 18.8 |
| 496 | 1 | 7 | 12.78 |

The empty driving distance between different parking yards is shown in Table 4. When the end of a task site is not the next start of a task site, there is an empty driving distance.

**Table 4.** Empty driving distance between different parking yards.

| | Lao Shan Bus Parking Yard | Si Hui Junction Station Parking Yard | Gu Cheng West Bridge Bus Parking Yard | Sun He Bus Parking Yard | Hui Zhong Li Parking Yard | Kang Jing Nan Li Parking Yard | National Stadium East Parking Yard |
|---|---|---|---|---|---|---|---|
| Lao Shan Bus Parking Yard | 0 | 37.7 | 1.9 | 44.7 | 29.8 | 42 | 30.2 |
| Si Hui Junction Station Parking Yard | 23.8 | 0 | 23.4 | 23.6 | 16.2 | 11 | 16.6 |
| Gu Cheng West Bridge Bus Parking Yard | 1.4 | 36.8 | 0 | 45.6 | 24 | 42.9 | 24.3 |
| Sun He Bus Parking Yard | 40.8 | 22.3 | 40.6 | 0 | 15.8 | 16.2 | 16.8 |
| Hui Zhong Li Parking Yard | 31.2 | 17.1 | 31.1 | 19.7 | 0 | 17.4 | 1.9 |
| Kang Jing Nan Li Parking Yard | 42.4 | 10.9 | 42.2 | 14.2 | 17.6 | 0 | 18.6 |
| National Stadium East Parking Yard | 22.4 | 19.1 | 21 | 24.4 | 3.5 | 21.6 | 3.9 |

The maximum mileage for the 12-meter-long electric buses is 133 km and 117 km for the 18-meter-long electric buses and the buses powered by fuel and natural gases have unlimited mileage. The quick charging time of pure electric buses is 30 min. The tasks of each route are executed jointly by the buses of seven parking yards. Convert the unit of energy consumption into the standard coal. According to the unit price of standard coal of 71.39 €/ton, $\omega_1 = 3.3$. The average annual salary level in Beijing in 2017 is 11033€. Calculated by 250 working days per year for 8 h per day, the average salary is 5.45 €/h, so $\omega_2 = \omega_3 = 5.45$. The parameters of the example are shown in Table 5.

**Table 5.** Setting of parameters in the calculation example.

| $\omega_1$ | $\omega_2$ | $\omega_3$ |
|---|---|---|
| 3.3 | 5.45 | 5.45 |

In the genetic algorithm proposed in this paper, the initial species size is 200 individuals, the crossover probability is 0.6, the mutation probability is 0.01, and the retention rate of good genes is 0.1.

*5.2. Results Analysis*

After calculation, the value of the objective function is 8456 € and a total of 144 buses are required. Among them, the number of vehicles needed for type V1, V2, V3 is 86, 14 and 44, respectively. The number of buses needed to be stopped in the parking yard is 48,36,19,16,8,7,10 respectively for the Si Hui Junction Station Parking Yard, Sun He Bus Parking Yard, Lao Shan Bus Parking Yard, Gu Cheng West Bridge Bus Parking Yard, Kang Jing Nan Li Parking Yard, Hui Zhong Li Parking Yard and National Stadium East Parking Yard. The total mileage of the tasks is 19,741 km and the total number of the charge is 70. The program iteration diagram is shown as Figure 4.

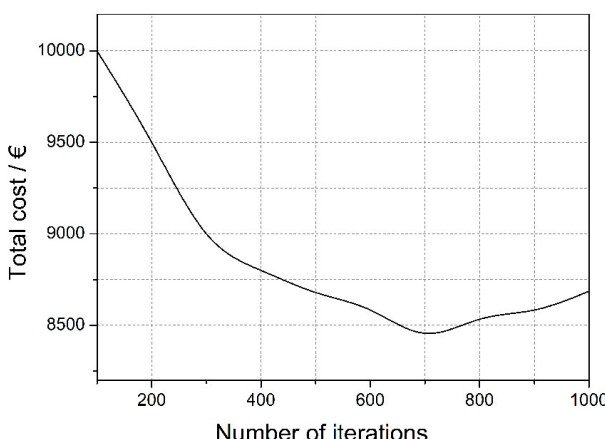

**Figure 4.** Program iteration diagram.

The bus order chain of each vehicle is shown in Table 6, in which 1041 to 1047 represents the parking yard one to seven, and if they are in the middle of the chain, it means the bus gets into the parking yard to charge.

**Table 6.** Bus order chain.

| Bus Sequence Number | Vehicle Type | Bus Order Chain |
| --- | --- | --- |
| 1 | V1 | 1041-682-429-759-934-38-882-47-893-1043 |
| 2 | V1 | 1043-411-1046-839-771-128-865-1045 |
| 3 | V3 | 1044-10-421-321-687-206-1047 |
| 4 | V1 | 1046-966-868-1041-236-257-727-203-354-763-1044 |
| 5 | V1 | 1042-326-163-1041-870-1043 |
| 6 | V1 | 1042-925-1046-545-379-1045 |
| 7 | V1 | 1042-511-1043-492-148-1046 |
| 8 | V1 | 1047-791-516-157-730-765-582-1043 |
| 9 | V3 | 1042-85-234-561-622-697-590-292-552-1041 |
| 10 | V3 | 1045-136-653-662-917-838-577-584-1046 |
| … | … | … |
| 142 | V2 | 1043-650-747-372-908-595-949-1044 |
| 143 | V3 | 1041-497-996-51-578-41-94-910-1047 |
| 144 | V1 | 1041-25-263-422-1046-651-846-931-1035-599-1046 |

## 5.3. Comparative Analysis

From the program iteration chart, we can see the variation trend. The optimal scheme is the lowest cost, but in the actual situation, it is impossible to achieve the optimal configuration at once. The capacity of the yard and the charging infrastructure need to be expanded. In the process of gradually replacing fuel vehicles by pure electric buses, we need to pay attention to how costs change and what proportion of fuel vehicles should be replaced by pure electric buses, so we choose different vehicle ratios to discuss. The cost of different plans with different vehicle type proportions is shown in Table 7. Energy consumption cost one refers to the energy consumption cost of pure electric buses, and energy consumption cost two refers to the energy consumption cost of diesel buses and liquefied natural gas buses. The proportion of two types of buses of five different plans and the changes in costs in each case are shown in Table 7.

**Table 7.** The cost of different vehicle type proportions.

| Plans | Number of Buses (Num) | | | Cost (Euro) | | | | Number of Charge Times |
|---|---|---|---|---|---|---|---|---|
| | Total Number of Buses | Number of Electric Buses | Number of Fuel Buses | Energy Consumption Cost1 | Energy Consumption Cost 2 | Passengers Cost | Total Cost | |
| Fuel buses increase by 20% | 132 | 59 | 73 | 2083 | 4860 | 1736 | 8679 | 46 |
| Fuel buses increase by 10% | 138 | 70 | 68 | 2142 | 4553 | 1888 | 8583 | 58 |
| The optimal plan | 144 | 83 | 61 | 3187 | 3187 | 2082 | 8456 | 70 |
| Fuel buses decrease by 10% | 150 | 95 | 55 | 3808 | 2539 | 2185 | 8533 | 82 |
| Fuel buses increase by 20% | 158 | 111 | 47 | 4080 | 2197 | 2306 | 8583 | 96 |
| Only electric buses | 205 | 205 | 0 | 6188 | 0 | 2238 | 8686 | 114 |

It can be concluded from the above table that:

1. From the 1–3 columns of the table, we can see that in the process of pure electric buses replacing fuel vehicles, the total number of vehicles increases gradually. The proportion of the number of reducing fuel vehicles and increasing pure electric buses is about 1:2., because pure electric buses with similar passenger capacity need to return to the parking lot to recharge after reaching the limited mileage. Therefore, in practical application, it is necessary to replace one fuel vehicle with two pure electric vehicles.

2. From the 4–5 columns of the table, we can see that energy consumption cost one increases while energy consumption cost two decreases. However, the energy consumption cost shows a decreasing trend, indicating that the energy consumption of the pure electric bus is much lower than that of the fuel vehicle. On the premise of ensuring the completion of the operation task, the energy consumption of the increased pure electric bus is lower than that of the reduced fuel vehicle. Therefore, considering the energy consumption alone, it is better to keep the higher the proportion of the electric bus.

3. The total cost is shown as concave in column 7 of Table 7, with decreasing energy costs and increasing passengers cost. It indicates that when the number of electric buses increases to a certain proportion, the saving costs on energy are not enough to make up for the rising passengers cost. There is a balance between the two costs in order to achieve the optimal (the lowest total cost). Therefore, it is not acceptable to merely consider energy consumption reduction from the perspective of the total cost. How to choose depends on which aspect the bus enterprises focus on, and at the same time, they should respond to the national policy.

4. As is shown in column 8 of Table 7, increasing charging times is inevitable in the process of increasing the proportion of pure electric buses, which requires the expansion of charging infrastructure.

5. The total number of vehicles in column 1 of Table 7 shows that in the process of replacing fuel buses by pure electric buses, the size of buses keeps increasing, so the capacity of the parking yard and charging equipment need to be expanded.

The current discussion on energy consumption saving is mostly concentrated in reducing the number of cars and formulating policies on reducing car use, but the studies on public transit are less. Based on the characteristics of energy-intensive for the traditional buses, blindly advocate public transit may save energy but it is not the best solution. The application of the electric bus in a reasonable ratio can achieve the reduction on public transport energy consumption. The current situation is that the environmental-friendly electric buses are gradually replacing the fuel vehicles. Although the replacement is not entire, the scale is gradually expanding. Therefore, the study on the schedule of the bus driving plan to reduce the energy consumption is particularly important. The variation of the single bus type operation is not reasonable, which may easily lead to the waste of capacity during flat peak periods and the shortage of capacity during peak periods. Different types of buses can carry different passenger loads, so different bus type matching ratios should be considered according to the passenger flow.

## 6. Conclusions

This paper makes an in-depth analysis on the relationship between the bus driving energy consumption and bus dispatching and bus type matching ratio under the background that pure electric buses gradually replaces traditional fuel buses, and many routes are operated with mixing pure electric buses and traditional fuel buses. There are mainly two steps to solve the problem. The first is to establish an optimization model of the multi-type bus operation energy consumption based on the time-space network and reduce the scale of the problem by cutting the empty driving arc. In the second, the genetic algorithm is applied to optimize the multi-objective function to obtain the optimal driving scheme. The proportion of electric vehicles replacing fuel vehicles is analyzed with examples

under the situation of gradual reduction of fuel vehicles. The optimal vehicle scheduling scheme and vehicle type ratio are obtained. In addition, the energy consumption cost and passenger cost are calculated in each case, and the suggestion of expanding the parking yards is given.

Currently, the Chinese government vigorously promotes the public transport with low energy consumption. The ratio of electric buses in the bus fleets increases, and mixing operation with multiple bus types makes the bus operating organization more complex. The paper innovatively takes the energy consumption as the objective and organizes the bus operation. The difference from the traditional fuel bus operating organization lies in the constraints of mileage range and charging time. In the multi-type energy consumption optimization model, the solution scale of the problem is reduced by establishing the time-space network and cutting the empty driving arcs. Due to the constraint of the charging time window of pure electric buses, two pure electric buses need to be added to replace one fuel bus, and the parking yard capacity needs to be expanded correspondingly. The bus type matching ratio is different for the situation considering the energy consumption cost alone and the situation considering the total cost. The decision depends on the preference of the decision makers. However, under the dual pressure of the environment and energy consumption, the growth of pure electric buses is a trend, and it also needs a stronger policy support from the government.

**Author Contributions:** Conceptualization, Y.L. and E.Y.; Methodology, Y.L. and E.Y.; Validation, E.Y.; Formal analysis, Y.L., S.L.; Writing—Original draft preparation, Y.L.; Writing—Review and editing, Y.L., E.Y., S.L.

**Funding:** This research was funded by National Key R&D Program of China, grant number 2018YFB1601300.

**Conflicts of Interest:** The authors declare no conflict of interest.

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
