# Peer review of "Energy Consumption Optimization Model of Multi-Type Bus Operating Organization Based on Time-Space Network"

_applsci, doi:10.3390/app9163352_

Round 1

Reviewer 1 Report

1. No explanation of some of the symbols used in the text.

2. If the article is dedicated to English-speaking readers, the currencies should be given in Euro.

3. It is not shown how the coding of the problem for the evolutionary algorithm occurred. How have the objective functions been adopted, what algorithm stop criteria, etc.?

4. Figure 2 is missing a population assessment block after the genetic operators have been triggered.

5. Lack of graphical depiction of the obtained test results

6. In chapter 5, several tables are presented without providing any explanation and text between them.

7.Why in this case coal is used as the reference unit of energy, not oil?

8. Which units apply in table 7

The article requires re-editing, the data are presented in a chaotic manner and not very understandable for the reader

Author Response

Dear Editors and Reviewers,
Thank you for your letter and for the reviewers’ comments and thoughtful suggestions on our manuscript entitled “Energy Consumption Optimization Model of Multi-type Bus Operating Organization Based on Time-Space Network” (Manuscript ID:applsci-560977). The comments are all valuable and very helpful for revising and improving our paper, as well as the important guiding significance to our researches. We have studied the comments carefully and have made correction which we hope meet with approval. Revised portions are marked in red in the paper. The responses to the reviewer’s comments are as flowing:

Point 1: No explanation of some of the symbols used in the text

Response 1: Thank you very much for pointing this out. According to this suggestion, we have supplemented some of the symbols used in the text in our revised paper. We have supplemented explanations of P, M, m, g, and. P represents bus order chain. M is the set of buses. m represents bus. g represents time period. is the minimum allowable full load rate of the research route, is the maximum allowable full load rate of the research route. At present, all symbols are explained in our revised paper.

Point 2: If the article is dedicated to English-speaking readers, the currencies should be given in Euro.

Response 2: Thank you very much for this helpful suggestion. According to this suggestion, we have modified currencies in Euro.

Point 3: It is not shown how the coding of the problem for the evolutionary algorithm occurred. How have the objective functions been adopted, what algorithm stop criteria, etc.

Response 3: Thank you very much for pointing this out. We adopts coding method of the whole vehicle. The designs of coding and coding scheme are shown in the first paragraph of chapter 4. A fitness calculation formula is added in chapter 4.2 to be more explicit in subsequent application. An individual selection formula and an individual selection step flow chart are supplemented to clarify how the individual is adopted in chapter 4.3.The termination principle is shown in chapter 4.4. Before reaching the maximum number of iterations, it is judged whether the average fitness of successive generations is unchanged or the change value is less than a minimum threshold. If so, the iteration process of the algorithm converges and the algorithm ends. Otherwise, if the maximum number of iterations has been reached, the new generation population obtained through selection, crossover and mutation will replace the previous generation population.

Point 4: Figure 2 is missing a population assessment block after the genetic operators have been triggered.

Response 4: Thank you very much for pointing this out. A group evaluation module is added after the fitness calculation in Figure 3(original Figure 2).

Point 5: Lack of graphical depiction of the obtained test results

Response 5: Thank you very much for pointing this out. A program iteration diagram is supplemented in revised paper. We have supplemented some depiction about it. At the same time, we have explained in detail about Table 7 to make the results more understandable.

Point 6: In chapter 5, several tables are presented without providing any explanation and text between them.

Response 6: Thank you very much for this suggestion. Some explanations as follows are supplemented between tables in chapter 5. Parking yard information is shown in Table 1.The number of Parking Space in Table 1 is one of the constraints which is calculated by formula (8). The vehicle information is shown in Table 2.The energy consumption per 100 km is applied in objective function (1). The specified passenger volume is one of the correlative conditions of the full load rate constraint formula (5). The bus route information is shown in Table 3.The route mileage is the key index for calculating energy consumption of objective function (1). The empty driving distance between different parking yards is shown in Table 4. When the end of a task site is not the next start of a task site, there is empty driving distance.

Point 7: Why in this case coal is used as the reference unit of energy,not oil?

Response7: Generally, standard coal is used as a unified unit for the measurement of energy consumption. Ton or kilogram of standard coal is the unified unit of measurement of energy in the world.. So we use coal as the reference unit of energy.

Point 8: Which units apply in table 7?

Response 8: Thank you very much for this suggestion. We have supplemented units in table 7.

At the same time, we have improved the manner that the data are presented. So, it will be more understandable for the reader.

We tried our best to improve the manuscript and made some changes in the manuscript.  These changes marked in red will not influence the framework of the paper. We appreciate for Editors/Reviewers’ warm work earnestly, and hope that the correction will meet with approval. Once again, thank you very much for your comments and suggestions. Wish you all the best!

Yours sincerely,

Shasha Liu, Enjian Yao, Shasha Liu

Email: [email protected]; [email protected]; [email protected]

August 8, 2019                              

Reviewer 2 Report

Dear Authors,

In my opinion, the paper is very interesting but must improve:

- the state of the art;

- add more references;

- line 143: what means "task"?

- fig1: what task can have a bus?

- line 291, from where you have the value between 4 and 60???

- table 4, if the buses have at the end of the route charging station, why the bus come back to charge?

- describe clear the values from table 3 and 4;

- table 7, explain in detail

Please improve your research

Author Response

Dear Editors and Reviewers,
Thank you for your letter and for the reviewers’ comments and thoughtful suggestions on our manuscript entitled “Energy Consumption Optimization Model of Multi-type Bus Operating Organization Based on Time-Space Network” (Manuscript ID:applsci-560977). The comments are all valuable and very helpful for revising and improving our paper, as well as the important guiding significance to our researches. We have studied the comments carefully and have made correction which we hope meet with approval. Revised portions are marked in red in the paper. The responses to the reviewer’s comments are as flowing:

Point 1: the state of the art

Response 1: At present, due to the constraints of endurance mileage, charging infrastructure, it is difficult to electrify all buses in a short period of time. There will inevitably be the mixed running of traditional fuel buses and pure electric buses. The current researches are hardly carried out under the background of mixing operation with multiple vehicle types such as electric buses and fuel buses, and there are no regional bus driving plan models considering both mileage range and charging time constraints. We conducts the research on regional pure electric bus driving plan with the objectives of energy consumption and bus service level, considering the pure electric bus charging constraints and mileage range constraints, and modify the algorithm to better support the application of pure electric buses in the real operation.  We have summarized the state in the last paragraph of chapter 1.

Point 2: Add more reference

Response 2: Thank you very much for this suggestion. According to this suggestion, we have supplemented twelve references (Number 5-10 and 20-25) in our revised paper.

Point 3:  Line 143:what means task ?

Response 3:  Thank you very much for pointing this out. In order to explain task arc, a method of generating task arc is added in the manuscript. Assuming that there are N task vehicles,  is the departure site of number i;  is the arrival site of number i;  is the departure time of number i;  is the arrival time of number i. According to the above process, the start node  can be formed, and the start node of the task can be arranged in chronological order to form a set of task start nodes T; the end node  can be arranged in chronological order to form a set of end-of-task nodes E. Task arc: (t, e), . The set of task arcs is.

Point 4: what task can have a bus

Response 4: In chapter 4.1, there is a provision that the first task is to randomly extract an electric vehicle from the parking lot for execution, and vehicles that execute the remaining task need to meet the following two requirements: 1. Remaining mileage subtracts possible empty-driving mileage is greater than task mileage adds the mileage returning to the nearest parking yard; 2. The time when last task was completed adds possible empty-driving time is less than the start time of the task adds charging time.

Point 5: In line 291, from where you have the value between 4 and 60?

Response 5: Thank you very much for pointing this out .Usually there are maximum and minimum departure interval constraints on bus routes. We refer to Beijing bus departure interval which is less than 5 min in peaking period and less than 15min in low peak period. We widen the constraints of departure frequency within a certain range, with 1 min and 15 min as upper and lower limits respectively.

Point 6: table 4, if the buses have at the end of the route charging station, why the bus come back to charge.

Response 6: Thank you very much for pointing this out. In the paper, after the completion of the task, the vehicle is in the terminal station, not the parking lot. There are two cases after the completion of the task: the first is that the vehicle will return to the parking lot from the terminal station and wait for the next task. The second is to drive directly from the terminal station to the starting station of the next task to perform the next task. But charging must be carried out in the parking lot, so when the electric vehicle is not enough to perform the next task, it must enter the parking lot to charge. There are many times in the article that return to the parking lot, which may cause misunderstanding. We have modified the terminology and Fig1 (b) in the manuscript.

Point 7: Describe clear values from table3 to table 4.

Response 7: Thank you very much for this helpful suggestion. We have supplemented the following contents in our revised paper. The bus route information is shown in Table 3.The route mileage is the key index for calculating energy consumption of objective function (1).The empty driving distance between different parking yards is shown in Table 4. When the end of a task site is not the next start of a task site, there is empty driving distance.

Point 8: table 7.explain in detail

Response 8: Thank you very much for this helpful suggestion. We have supplemented the comparison scheme decription, described the results in the table and given some suggestions combined with the actual work of bus operation.

We tried our best to improve the manuscript and made some changes in the manuscript.  These changes marked in red will not influence the framework of the paper. We appreciate for Editors/Reviewers’ warm work earnestly, and hope that the correction will meet with approval. Once again, thank you very much for your comments and suggestions. Wish you all the best!

Yours sincerely,

Yuhuan Liu, Enjian Yao, Shasha Liu

Email: [email protected]; [email protected]; [email protected]

August 8, 2019  

Round 2

Reviewer 1 Report

Thank you for the changes and explanations given.

Best Regards

Author Response

Dear Editors and Reviewers,

Thank you very much for your comments and suggestions. Wish you all the best!

Yours sincerely,

Yuhuan Liu, Enjian Yao, Shasha Liu

Email: [email protected]; [email protected]; [email protected]

August 12, 2019

Reviewer 2 Report

The authors have sufficiently improved and corrected the manuscript.

In a figure1 b , line 197 are china words, please change in english

Author Response

Dear Editors and Reviewers,
Thank you for your letter and for the reviewers’ comments and suggestions on our manuscript entitled “Energy Consumption Optimization Model of Multi-type Bus Operating Organization Based on Time-Space Network” (Manuscript ID:applsci-560977). The comment is valuable and very helpful for revising and improving our paper. We have studied the comments carefully and have made correction which we hope meet with approval. Revised portions are marked in red in the paper. The responses to the reviewer’s comments are as flowing:

Point 1: In a figure1 b, line 197 are china words, please change in English

Response 1: Thank you very much for pointing this out. We have changed it in English.

We tried our best to improve the manuscript and made some changes in the manuscript. We appreciate for Editors/Reviewers’ warm work earnestly, and hope that the correction will meet with approval. Once again, thank you very much for your comments and suggestions. Wish you all the best!

Yours sincerely,

Yuhuan Liu, Enjian Yao, Shasha Liu

Email: [email protected]; [email protected]; [email protected]

August 12, 2019
